# Expression of IL-37 Induces a Regulatory T-Cell-like Phenotype and Function in Jurkat Cells

**DOI:** 10.3390/cells11162565

**Published:** 2022-08-18

**Authors:** Douglas Grant Osborne, Joanne Domenico, Mayumi Fujita

**Affiliations:** 1Department of Dermatology, University of Colorado Anschutz Medical Campus, Aurora, CO 80045, USA; 2Department of Immunology and Microbiology, University of Colorado Anschutz Medical Campus, Aurora, CO 80045, USA; 3Department of Veterans Affairs Medical Center, VA Eastern Colorado Health Care System, Aurora, CO 80045, USA

**Keywords:** IL-37, regulatory T cell, Treg, FOXP3, Jurkat

## Abstract

The anti-inflammatory cytokine interleukin-37 (IL-37) plays a key role in inhibiting innate and adaptive immunity. Past results have shown that IL-37 is elevated in human Treg cells compared to other T cell subsets and contributes to enhancing the Treg transcription factor, forkhead box protein P3 (FOXP3). However, it is unknown if ectopic expression of IL-37 in non-Treg CD4+ T cells can lead to the development of Treg phenotype and function. In the present study, we used a PrimeFlow^®^ RNA assay and confirmed elevated *IL37* expression in human Treg cells. We then stably transfected the non-Treg CD4+ T cell leukemia cell line, E6 Jurkat cells, with *IL37* and found significant induction of the Treg phenotype. These IL-37-expressing Jurkat cells had elevated CTLA-4 and FOXP3 and produced IL-10. In conjunction with the Treg phenotype, IL-37-expressing Jurkat cells suppressed T cell activation/proliferation, comparable to human primary Treg cells. The creation of this stable human Treg-like cell line has the potential to provide further assistance for in vitro studies of human Treg cells, as it is more convenient than the use of primary human Treg cells. Furthermore, it provides insights into Treg cell biology and function.

## 1. Introduction

Regulatory T (Treg) cells play a critical role in peripheral tolerance by suppressing and preventing an auto-reactive immune response [1]. Understanding how Treg cells function in maintaining peripheral tolerance is key to preventing and treating autoimmunity [2] and for the future development of immune-targeted therapies, such as adoptive Treg cell therapy [3]. However, culturing and maintaining sufficient primary human Treg cells to perform molecular-based experiments is time-consuming and expensive. While novel genome editing techniques and the identification of stable Treg markers, such as GPA33, have improved the efficacy of modifying primary Treg cells [4,5], experimental challenges still exist due to many factors, including the inter-personal and intra-personal heterogeneity of primary human Treg cells [4]. The alternative to primary Treg cells is to develop Treg-like cell lines or cells easily usable for experiments or therapies. Multiple studies have attempted to identify or establish Treg-like cell lines or cells by altering gene expression or inducing Treg cell differentiation in non-Treg cell lines [3,6] but have yet to produce an artificial Treg cell line that is either stable or readily available for use.

Developing a Treg-like cell line requires stable expression of the transcription factor forkhead box P3 (FOXP3), a key factor used to identify Treg cells and required for their suppressive capabilities [5]. In 2007, Kim et al. transfected the non-Treg CD4+ T cell leukemia cell line, E6 Jurkat cells, with a FOXP3 overexpression vector to create a Treg-like cell line [6]. They found that overexpression of FOXP3 in Jurkat cells led to an elevation in surface expression of cytotoxic T-lymphocyte antigen 4 (CTLA-4) and CD25. While these cells suppressed the proliferation of conventional T (Tconv) cells, the suppressive activity was significantly lower compared to primary Treg cells [6], suggesting that alternative pathways are required to create functionally comparable artificial Treg cells from Jurkat cells.

Recently, the anti-inflammatory IL-1 cytokine family member IL-37 was shown to contribute to the expression of FOXP3 [7,8]. Knockdown of *IL37* in primary human Treg cells resulted in a significant decrease in FOXP3 and promoted T cell proliferation and differentiation [7]. Treatment of human Treg cells with recombinant IL-37 also upregulated FOXP3 and enhanced their suppressive activity [8]. These studies indicate a critical role of IL-37 in human Treg cells.

However, it has been unclear whether T cells, including CD4+ Treg cells, express IL-37. Rudloff et al. reported that monocytes and dendritic cells are the main producers of IL-37 in human peripheral blood mononuclear cells (PBMCs) from healthy donors [9]. They found that only 0.5% of T cells expressed IL-37. On the other hand, we reported that *IL37* mRNA and IL-37 protein were highly expressed in human T cells, including Treg cells from healthy volunteers [10]. Because of the contradictory findings, the present study employed another method, a PrimeFlow^®^ RNA assay, to examine *IL37* expression in human PBMCs and studied the role of IL-37 expression in T cells.

We confirmed that human Treg cells expressed elevated *IL37*. On the other hand, IL-37 expression was minimal or undetectable in the non-Treg CD4+ T cell leukemia cell line, Jurkat cells, prompting us to use Jurkat cells to study biological changes after IL-37 overexpression. We demonstrated the induction of Treg-like phenotype and function by stably transducing Jurkat cells with lentivirus made from an IL-37 overexpressing vector (*IL37* OE). Jurkat cells are suitable for developing a Treg-like cell line due to their CD4+ T cell phenotype, ease of culturing and genetic manipulation, and ability to become suppressive [6]. Our results provide an alternative method for developing a stable Treg-like cell line.

## 2. Materials and Methods

### 2.1. Cell Preparation and Culture

Human leukemia cell line E6 Jurkat cells were obtained from ATCC (Manassas, VA, USA). All cells were cultured in RPMI 1640 with L-glutamine (Gibco, ThermoFisher, Waltham, MA, USA) and supplemented with 10% FBS (Gemini BioProducts, CA, USA), 10 nM HEPES (Life Technologies, ThermoFisher, Waltham, MA, USA), 1% non-essential amino acids (Life Technologies, ThermoFisher), and 1 mM sodium pyruvate (Life Technologies, ThermoFisher) in a humified incubator at 37 °C and 5% CO_2_. Cells were cultured at 2 × 10^5^/mL in a Nunc^TM^ EasYFlask T75 (Sigma, St. Louis, MO, USA), passaged when cell counts reached 1 × 10^6^/mL, and reseeded at 1:5 dilution in a new flask.

### 2.2. Antibodies

Flow cytometry and cell stimulation utilized the following antibodies from Thermo Fisher (Waltham, MA, USA) directed against antigens: IL-10 (PE-Cy7, JES3-9D7), CD28 (unconjugated, CD28.2), CD3 (unconjugated, OKT3), CTLA-4 (PE, 14D3), IL-37 (PE, 37D12), CD127 (eFluor450, eBioRDR5), and FOXP3 (eFluor450, D608C).

### 2.3. PrimeFlow^®^ RNA Assay

Human donor PBMCs were harvested and purified using Ficoll gradient centrifugation. Cells were then either left untreated or cultured with 100 ng/mL lipopolysaccharide (LPS) in complete RPMI 1640 media or resuspended in freeze media (RPMI + 10%DMSO (Sigma) and placed in a −80 °C freezer for later use. After 24 h, fluorescent mRNA in situ hybridization was performed with the PrimeFlow^®^ RNA Assay from Thermo Fisher with custom-designed *IL37* mRNA-specific probes according to the manufacturer’s instructions. The *IL37* mRNA probe was designed by Thermo Fisher based on our published *IL37* qRT-PCR primer sequences [10] to obtain consistent results between the two different assays. We also used the same sequences for the qRT-PCR of this paper, shown in Table 1. For surface staining, the manufacturer’s recommended amount (5 μL/test) of fluorescently conjugated antibody specific to the surface marker was added to each donor PMBC sample seeded at 1 × 10^6^ cells and resuspended in 100 μL of FACS buffer for 30 min at 4 °C, then washed twice with FACS buffer prior to being either analyzed or further processed. Immune cells were stained into two groups based on the manufacturer’s provided immune antibody panels, one set for myeloid subsets and the other for lymphoid subsets (shown by the gating strategies in Figure 1A,B). Once gated on the individual subset, cells were analyzed for the expression of *IL37* mRNA (AF437). The PrimeFlow^®^-specific antibodies for cell surface proteins, including CD3 (efluor450, UCHT1), CD4 (AF700, RPA-T4), CD8 (APC-efluor780, RPA-T8), CD11c (PE-efluor610, 3.9), CD14 (PECy7, 61D3), CD16 (PE, EBIOCB16), CD19 (PECy5.5, SJ25C1), CD25 (PE-Cy7, BC96), CD45RB (AF488, PD7/26), CD56 (PE-Ef610, CMSSB), and HLA-DR (FITC, L243), were purchased from Thermo Fisher. Fixable Viability Dye (eflour506) was used to eliminate dead cells. Flow cytometry of PrimeFlow^®^ RNA samples was performed on a ZE5 Cell Analyzer (BioRad, Hercules, CA, USA). Further analysis of flow cytometry data was conducted using FlowJo_V10 software (Tree Star, Ashland, OR, USA).

### 2.4. Jurkat E6 Transduction

Jurkat cells were transduced with lentivirus made from transfecting 293T cells transfected with either an empty vector or pLenti-IL37-C-Myc-DDK-P2A-Puro vector (*IL37* OE) (Origene, Rockville, MD, USA) together with the Lenti-V-pak packaging kit (Origene, MD, USA). Supernatants containing lentiviral particles were harvested 48 h after transfection. The presence of lentivirus in the supernatants was confirmed using Lenti-X GoStix Plus (Takara, San Jose, CA, USA) according to the manufacturer’s instructions. An amount of 1 × 10^6^ Jurkat cells were transduced using 1 mL lentivirus-containing supernatants and 8 μg/mL polybrene and incubated for 24 h at 37 °C. Media was then changed to RPMI 1640 supplemented with pen-strep and 10% FCS. Drug selection was started 48 h after transduction with 0.5 μg/mL puromycin. Jurkat cells were bulk selected, and overexpression was confirmed by Western blot.

### 2.5. Flow Cytometry: Surface and Intracellular Antigen Staining, Activation, and Proliferation

Intracellular FOXP3 and cytokines were assayed by flow cytometry, as described [11]. Before fix/permeabilization, cells were surface-stained at 4 °C for 30 min. Purified lymphocytes were washed with FACS buffer (5% BSA (Sigma-Aldrich, Burlington, MA, USA) and 1× PBS) and then prepared for surface or intracellular staining, as described [11]. For surface staining, the manufacturer’s recommended amount of fluorescently conjugated antibody specific to the surface marker to be visualized was added to 1 × 10^6^ cells resuspended in 100 μL of FACS buffer for 30 min at 4 °C, then washed twice with FACS buffer prior to being either analyzed or further processed. In the latter, following twice wash with FACS buffer, cells were immediately fixed in 4% paraformaldehyde (PFA) and permeabilized with 90% methanol, followed by staining. For IL-10 intracellular cytokine staining, the FOXP3 Cytoperm/Cytofix staining kit (BD Pharmingen, San Diego, CA, USA) was used after cells were stimulated for 24 h on 1 μg/mL anti-CD3-coated plates with soluble 1 μg/mL anti-CD28 antibody or anti-CD3/CD28 conjugated Dynabeads (Gibco, Thermo Fisher) and treated in culture for 6 h with BD GolgiStop^TM^ (BD, Franklin Lakes, NJ, USA). T cell proliferation was measured by staining cells with 5 μM CFSE (Molecular Probes, Eugene, OR, USA) before cell culture and analyzing them using flow cytometry.

### 2.6. RNA Extraction and Quantitative RT-PCR Analysis

As described above, cells were stimulated for 24 h on anti-CD3/CD28-coated plates. RNA was extracted from Jurkat cells using the RNeasy Plus Mini Kit (Qiagen, Germantown, MD, USA) and subsequently reverse-transcribed using the iScript cDNA Synthesis Kit (Bio-Rad, CA, USA). Quantitative RT-PCR (qRT-PCR) was performed with Power Up SYBR Green PCR Master Mix (Applied Biosystems, Foster City, CA, USA) on the AriaMx Real-Time PCR system (Agilent Technologies, Santa Clara, CA, USA). The primer sequences used for human cells are shown in Table 1. Please note that the primer sequences for *IL37* are the same as those used for the PrimeFlow^®^ RNA assay. Quantification of mRNA was measured using change-in-cycling-threshold (ΔCt) and is shown as the ratio relative to *GAPDH* mRNA expression.

### 2.7. Western Blot

Western blot analysis was performed as previously described [12]. Rabbit anti-human IL-37 (IL-1F7) polyclonal antibody (Invitrogen, ThermoFisher, MA, USA) at a dilution of 1:1000 was used for detection of IL-37, and mouse β-actin monoclonal antibody (Sigma) at a dilution of 1:10,000 was used for the loading control). HRP-conjugated anti-rabbit (1:5000) and anti-mouse (1:5000) IgG were purchased from Sigma-Aldrich. The SuperSignal West Pico Kit (Thermo Fisher) was used, according to the manufacturer’s instructions, for chemiluminescent detection of the proteins.

### 2.8. In Vitro T Cell Suppression Assay

Human CD4+CD127loCD25+ T cell Isolation Kit (Miltenyi, Auburn, CA, USA) was used to isolate human CD4+CD25^lo^ T cell responders (Tresp) and human CD4+CD25^hi^ T cells (human primary Treg cells, positive control) for suppression assays [13]. An amount of 100,000 purified CFSE-labeled Tresp in 50 μL of complete RPMI 1640 were added to a round-bottom 96-well plate (CellStar, Dallas, TX, USA) containing heat-killed splenocytes (as the source for antigen-presenting cells), 5 μg/mL soluble anti-CD3 antibody, and the indicated ratio of Jurkat/Treg cells all in a 150 μL of complete RPMI 1640 [14]. A total of 5 days later, CFSE was used to measure proliferation/suppression of Tresp cells and analyzed by flow cytometry. Percent suppression was calculated using the equation, % suppression = 100 − (X/Y)100, X represents the division index (DI) of Tresp with Jurkat/Treg cells, and Y is the average DI of Tresp cells alone, as in [15]. 

### 2.9. IL-37 ELISA

IL-37 secretion into the supernatant was analyzed by culturing 1 × 10^6^ vector control or *IL37* OE Jurkat cells in 1 mL of complete RPMI 1640 medium. Supernatants were then collected and analyzed using DuoSet^®^ human IL-37/IL-1F7 ELISA kits (R&D Systems, Minnneapolis, MN, USA) to measure IL-37 protein abundance, according to the manufacturer’s instructions. As controls, 1 and 2 ng/mL recombinant human IL-37 (R&D) were used to measure the sensitivity of the ELISA assay. 

### 2.10. Statistical Analysis

Data are expressed throughout as mean ± standard error mean (SEM). Data sets were compared using the two-tailed unpaired Student’s *t*-test. Statistical analysis (Student’s *t*-test, ANOVAs, and column statistics) and graphing were performed using Prism 4 (GraphPad, La Jolla, CA, USA). Differences were considered statistically significant when *p* < 0.05.

## 3. Results

### 3.1. IL37 Expression Is Elevated in Human Treg Cells

We used a flow cytometry-based assay to measure *IL37* mRNA levels in freshly obtained human PBMCs without culture or fractionation. The PrimeFlow^®^ assay was used to simultaneously measure the gene expression of a specific target, *IL37*, in multiple immune cell subsets and provide single-cell analyses [16]. Cells were stimulated with LPS, similar to the condition from another study to induce IL-37 [9]. The gating strategy in Figure 1A (myeloid) and B (lymphoid cells) was performed to measure the quantity of *IL37* in multiple immune cell types (Figure 1C). As shown in Figure 1D, the majority of monocytes (87%) and Treg cells (CD4+CD25^hi^CD45RB^lo^) (80%) expressed *IL37* mRNA, and their frequencies increased further from 87% to 97% in monocytes and 80% to 91% in Treg cells after LPS stimulation. *IL37* was expressed in more than 50% of other T cell populations (CD3+, CD8+, CD4+, and Tconv cells), and their frequencies increased after LPS stimulation. On the other hand, *IL37* was expressed in only a portion of other populations (dendritic cells, B cells, and NK cells), and their frequencies were not changed after LPS stimulation.

We further analyzed *IL37* expression levels in each cell type (Figure 1E). The *IL37* mRNA level was highest in monocytes compared to other immune subsets, and the level further increased after LPS stimulation. Treg cells expressed the highest *IL37* level among lymphoid subsets, which further increased after LPS stimulation. Based on these results and the results of others [9,10], we conclude that *IL37* is highly expressed in human Treg cells. Considering its role in inducing FOXP3, we speculated that IL-37 would be a good candidate for establishing a Treg-like cell line from non-Treg CD4+ cells expressing lower IL-37.

### 3.2. Overexpression of IL-37 Leads to the Development of a Treg Phenotype in Jurkat Cells

To develop a stable cell line expressing IL-37, we generated a lentiviral FLAG-tag *IL37* overexpressing vector (vector shown in Figure 2A). To measure *IL37* mRNA, we employed qRT-PCR using the same primer sets used for the PrimeFlow^®^ RNA assay. Human leukemia E6 Jurkat cells and their empty vector-transduced cells express very little IL-37 at both the gene (Figure 2B) and protein (Figure 2C) levels, making them suitable for studying IL-37-induced changes. When FLAG-tag IL37 was transduced, the Jurkat cells overexpressed IL-37 at the gene (~5.8-fold increase) and protein (~*9.2*-fold increase) levels, confirming successful transduction and translation, respectively (Figure 2B,C). *IL37* overexpressing (OE) Jurkat cells had elevated *FOXP3* gene expression compared to control Jurkat cells (Figure 2D), which was further confirmed at the protein levels by Western blotting (Figure 2C) and flow cytometry (Figure 2E). Specifically, FOXP3 densitometry measurements showed a 78-fold increase in FOXP3 protein expression in *IL37* OE Jurkat cells.

Along with elevated FOXP3, *IL37* OE Jurkat cells showed elevated gene expression of *CTLA4*, *IL10*, glucocorticoid-induced TNFR-related (*GITR*), and transforming growth factor-β (*TGFβ*) (Figure 3A), markers commonly found on highly suppressive Treg cells [3]. We confirmed elevated CTLA-4 surface protein expression by flow cytometry (Figure 3B). Furthermore, a significant proportion of *IL37* OE Jurkat cells expressed IL-10 protein without stimulation, which was further increased with CD3/CD28 stimulation compared to control Jurkat cells (Figure 3C).

These data confirm that IL-37 overexpression in Jurkat cells induced a Treg cell phenotype. It is known that both CTLA-4 and IL-10 play a significant role in suppressing an adaptive immune response [17,18,19], suggesting that IL-37 overexpression in Jurkat cells also induces Treg functionality.

### 3.3. IL37 OE Jurkat Cells Suppress Naïve CD4+ T Cell Proliferation Comparable to Human Primary Treg Cells

Since FOXP3 expression is integral in Treg suppressive function, we examined if *IL37* OE Jurkat cells exhibited suppressive function in vitro. Freshly isolated human CD4+CD25^lo^ T cell responder (Tresp) cells were labeled with CFSE and co-cultured in CD3-stimulatory conditions for 5 days with either control Jurkat cells or *IL37* OE Jurkat cells. Human primary Treg cells were used as a positive control. We found that the co-culture of Tresp cells with *IL37* OE Jurkat cells significantly suppressed the proliferation of Tresp cells (~50% reduction compared to control Jurkat cells), comparable to human primary Treg cells (Figure 4). No significant difference was found in suppressive ability between *IL37* OE Jurkat cells and human primary Treg cells across all Treg/Tresp ratios.

Because secreted/extracellular IL-37 has been shown to inhibit cell activation [8,20,21], we examined if IL-37 was released from *IL37* OE Jurkat cells to suppress Tresp cells. IL-37 ELISA showed that the secretion of IL-37 from *IL37* OE Jurkat cells was minimal and compatible with the level of IL-37 secretion from the vector control (Appendix A), suggesting that intracellular IL-37 expression, but not extracellular IL-37 secretion, was responsible for the suppressive function of *IL37* OE Jurkat cells.

Together, these data confirm that stable IL-37 expression in Jurkat cells induced a Treg-like phenotype and function. Our results provide an alternative method for developing of Treg-like cell line by overexpressing IL-37 in non-Treg CD4+ T cells.

## 4. Discussion

In this study, we used a flow cytometry-based gene expression assay and confirmed our earlier published results that human primary Treg cells expressed an elevated level of IL-37 [10]. We also demonstrated that the stable overexpression of IL-37 in a non-Treg CD4+ T cell line, E6 Jurkat cells, induced a Treg-like phenotype (elevated FOXP3, CTLA-4, IL-10, and other Treg markers) and suppressive function.

IL-37 has been extensively studied in innate immunity and inflammatory conditions [22]. However, its role in adaptive immunity remains less understood. While multiple studies reported the tolerogenic role of IL-37 in dendritic cells [21,22], only a handful of studies have investigated IL-37 in lymphocytes [8,9,10,20]. A possible explanation for the limited studies in lymphocyte subsets could be related to the scarcity of information about IL-37’s expression in lymphocytes. The current study used a PrimeFlow^®^ RNA assay and demonstrated that *IL37* was elevated in human Treg cells. While Rudloff et al. reported that only 0.5% of T cells expressed IL-37 [9], their study and our current study use different assays (measuring protein vs. RNA) and different gating strategies (gating IL-37-positive cells first vs. gating each cell subset first to analyze *IL37* expression), thus preventing in-depth comparison of the two studies. However, our current results using an *IL37* mRNA probe are consistent with our previous results using the IL-37 monoclonal antibody (37D12) [10], confirming a high expression of IL-37 in human Treg cells.

We also found that the expression levels of *IL37* further increased in monocytes and lymphocytes with LPS stimulation. IL-37 is known to be induced or increased by many stimuli, including LPS in macrophages [23]. As LPS activates cells, we wondered if the observed increase in MFI was due to the increased background staining from enlarged cells. We measured cell sizes following LPS stimulation and observed only a slight increase in monocyte and dendritic cells but no size changes in T cells after LPS stimulation (Appendix A). A PrimeFlow^®^ RNA assay uses fixation and permeabilization (fix/perm) for nuclear staining, reducing cell sizes even after cell activation [24]. These data confirm that increased *IL37* expression was not due to the increased background staining influenced by cell sizes.

Treg cells are essential in peripheral tolerance to suppress and prevent an auto-reactive immune response [1]. Stable expression of FOXP3 in Treg cells is required for their suppressive capabilities and stability and is critical in maintaining peripheral tolerance [5]. Because Jurkat cells expressed almost undetectable levels of IL-37, we speculated that overexpressing IL-37 could induce FOXP3 expression and suppressive capacity in Jurkat cells. We investigated the function of *IL37* OE Jurkat cells and found that they suppressed Tresp proliferation comparable to the primary human Treg cells. In comparison, Kim et al. reported that the suppressive function of *FOXP3* OE Jurkat cells was lower than primary human Treg cells [6]. While we cannot directly compare their *FOXP*3 OE Jurkat cells and our *IL37* OE Jurkat cells due to different expression systems and experimental conditions, the difference in suppressive function may derive from mechanistic differences. Since IL-37 interacts with the FOXP3 transcription factor SMAD3 [25], IL-37 could contribute to *FOXP3* gene expression via SMAD3. SMAD3 represses IFN-γ and inhibits T-BET, the Th1 transcription factor [26] expressed in Jurkat cells. Therefore, IL-37 expression in Jurkat cells might alter the T-cell differentiation balance through Th1/Treg cell transcription factors via SMAD3. Further analyses are needed to compare *IL37* OE Jurkat cells to primary human Treg cells and examine signaling pathways to maintain the Treg phenotype and function.

Understanding the mechanism of Treg plasticity and how Treg cells can lose FOXP3 and convert to a pro-inflammatory CD4+ T cell subset [27] is crucial to developing successful immunotherapies, such as Treg adoptive transfer therapy [3]. Creating a stable Treg cell line, such as with our *IL37* OE Jurkat cells, will provide a convenient model for analyzing Treg plasticity, helping to understand the molecular mechanisms that support FOXP3 expression and aiding in the development of therapeutics for autoimmunity and cancer.

## 5. Conclusions

In this study, we confirmed elevated *IL37* expression in human Treg cells. We transduced the non-Treg CD4+ T cell leukemia cell line, E6 Jurkat cells, with lentivirus made from an *IL37* overexpressing vector (*IL37* OE) and demonstrated the induction of Treg-like phenotype and function in *IL37* OE Jurkat cells, compatible to human primary Treg cells. Jurkat cells are suitable for developing a Treg-like cell line due to their CD4+ T cell phenotype, ease of culturing and genetic manipulation, and ability to become suppressive. Our results provide an alternative mechanism for developing a stable Treg-like cell line that can be used for studying Treg cell biology and function.

## Figures and Tables

**Figure 1 cells-11-02565-f001:**
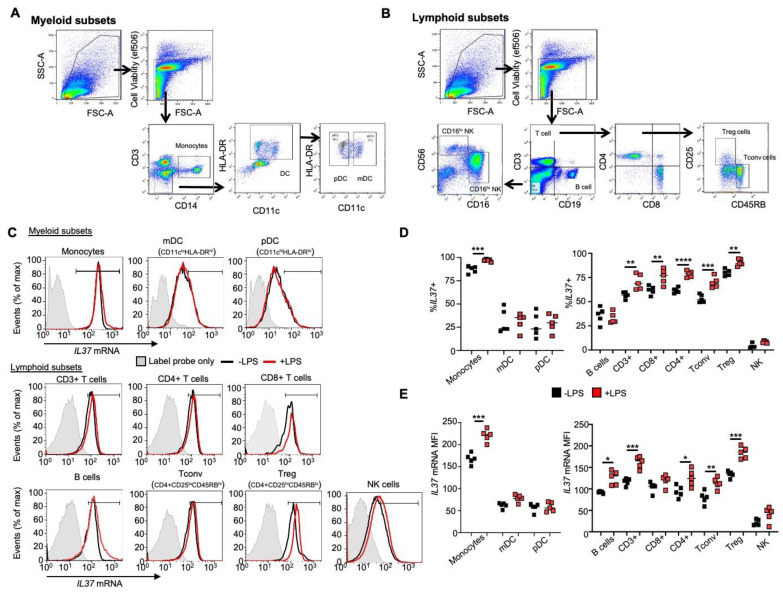
Identification of *IL37*-expressing cells in human PBMCs using a flow cytometry-based PrimeFlow^®^ RNA assay. (**A**,**B**) Myeloid (**A**) and lymphoid cell (**B**) gating strategy analyzing the expression of cell surface proteins and *IL37* mRNA from immune cells. (**C**–**E**) Histogram (**C**) and quantification by frequency (**D**) and mean fluorescent intensity (MFI) (**E**) of *IL37* mRNA expression in myeloid and lymphoid cell subsets. PBMCs were treated in the absence (-), black lines or black symbols) or presence (+, red lines or red symbols) of 100 ng/mL LPS for 24 h. A gray histogram represents the fluorescence of the label probe only (background) for each immune cell subset as a negative control. Gating for determining positive cell frequency was established to exclude ~99% of the control events as negative. *RPL13a* mRNA served as an internal control (not shown). Each symbol represents an individual donor; small horizontal lines indicate mean ± s.e.m., (n = 5 donors per group) (**A**–**E**). * *p* < 0.05; ** *p* < 0.01; *** *p* < 0.001; **** *p* < 0.001 (Student’s *t*-test). Data are representative of two independent experiments.

**Figure 2 cells-11-02565-f002:**
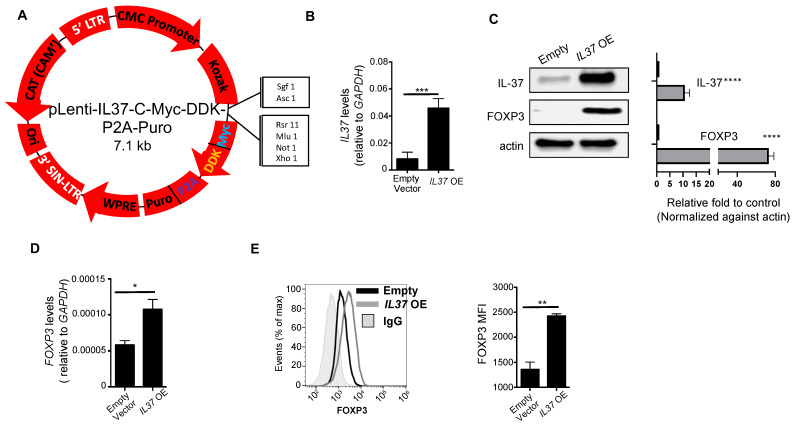
IL37 overexpression induces FOXP3 in Jurkat cells. (**A**) pLenti-IL37-C-Myc-DDK-P2A-Puro vector (*IL37* OE) (Origene). (**B**) qRT-PCR analysis of *IL37* mRNA expression in Jurkat cells transfected with an empty vector or *IL37* OE. *GAPDH* served as an internal control, and the values represent *IL37* gene expression as a ratio to *GAPDH* expression. (**C**) IL-37 and FOXP3 immunoblotting of empty vector and *IL37* OE Jurkat cells. Actin was used as a loading control. Representative immunoblotting (left panels) and quantification of bands (right panels) of three immunoblot experiments. The band densities of proteins were quantified with Image J, normalized to actin, and expressed as fold changes compared to control Jurkat cells expressing an empty vector. (**D**) *FOXP3* mRNA expression in vector control and *IL37* OE Jurkat cells. *GAPDH* served as an internal control, and the values represent *FOXP3* gene expression as a ratio to *GAPDH* expression. (**E**) Flow cytometry analyses of FOXP3 protein expression in vector control and *IL37* OE Jurkat cells. Histograms (top) and quantification by mean fluorescent intensity (MFI) (bottom) of FOXP3 expression. IgG used for negative control staining (solid gray histogram). Small horizontal lines indicate mean ± s.e.m. * *p* < 0.05; ** *p* < 0.01; *** *p* < 0.001; **** *p* < 0.001 (Student’s *t* test). Data are representative of three independent experiments.

**Figure 3 cells-11-02565-f003:**
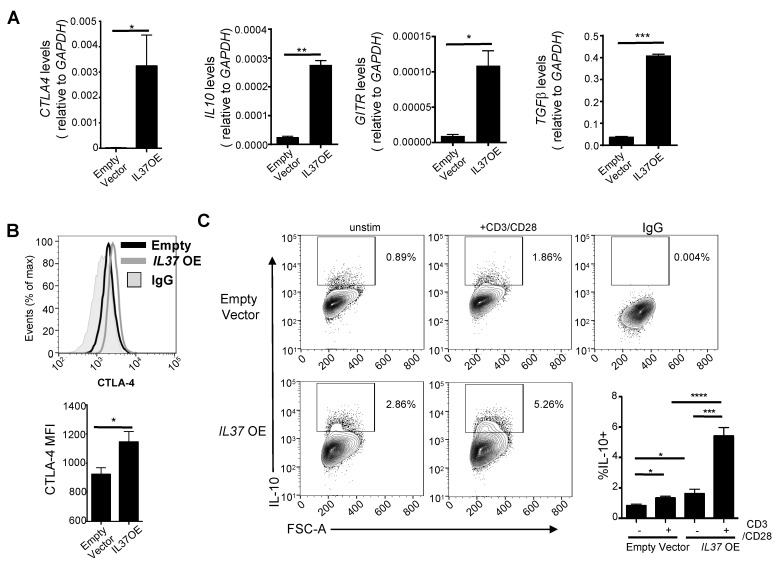
IL-37 overexpressing Jurkat cells are phenotypically similar to Treg cells. (**A**) qRT-PCR analysis of *CTLA4, IL10, GITR*, and *TGFB* mRNA expression in vector control and *IL37* OE Jurkat cells. *GAPDH* served as an internal control, and the values represent the gene expression level as a ratio to *GAPDH* expression. (**B**) CTLA-4 protein expression in vector control and *IL37* OE Jurkat cells. Histograms (upper) and quantification by mean fluorescent intensity (MFI) (lower) of CTLA-4 expression. IgG was used for negative control staining (solid gray histogram). (**C**) Left, contour plots of IL-10+ cell gating in unstimulated and anti-CD3/CD28 treated vector control and *IL37* OE Jurkat cells. Right lower, quantification of the percentage of IL-10+ Jurkat cells. IgG was used for gating for positive staining. Small horizontal lines indicate mean ± s.e.m. * *p* < 0.05; ** *p* < 0.01; *** *p* < 0.001; **** *p* < 0.0001 (Student’s *t* test). Data are representative of four independent experiments.

**Figure 4 cells-11-02565-f004:**
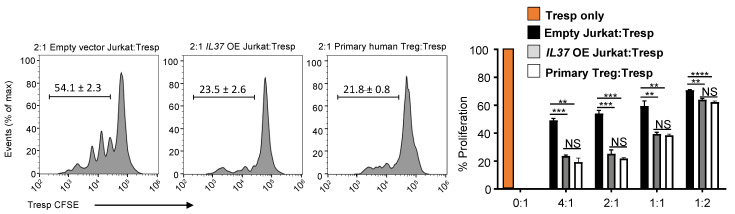
IL-37 overexpressing Jurkat cells are highly suppressive. In vitro T cell suppression assay by vector control and *IL37* OE Jurkat cells. Histograms (left) and % proliferation (right) showing division of CFSE-labeled CD4+CD25- T cell responders (Tresp) purified from healthy human donor PBMCs, cultured with anti-CD3 and either vector control Jurkat cells (black bars), *IL37* OE Jurkat cells (gray bars), or human primary Treg cells (white bars) at Treg:Tresp ratios of 4:1 to 1:2 for 5 days. Purified Tresp cells without Jurkat cells or Treg cells (0:1) were used as a positive control of Tresp proliferation without suppression (an orange bar). Data represent mean ± s.e.m. NS, not significant (*p* > 0.05); ** *p* < 0.01; *** *p* < 0.001; **** *p* < 0.0001 (Student’s *t* test). Data are representative of two independent experiments.

**Table 1 cells-11-02565-t001:** Primer set sequences.

Gene	Primer Sets (5′-3′)
*CTLA4*	F-CTCTACATCTGCAAGGTGGAGC R-AGAGGAGGAAGTCAGAATCTGGG
*FOXP3*	F-CAGAGCTCCTACCCACTGCT R-CTTCTCCTTCTCCAGCACCA
*GAPDH*	F-TGCACCACCAACTGCTTAGC R-GGCATGGACTGTGGTCATGAG
*GITR*	F-CATGTGTGTCCAGCCTGAAT R-GGCACAGTCGATACACTGGA
*IL37*	F-GCATTCATGACCAGGATCAC R-CAAAGAAGATCTCTGGGCGTA
*TGFB*	F-CACCTGGAGCTGTACCAGAA R-TGCAGTGTGTTATCCCTGCT

## Data Availability

All raw data supporting the present study results can be obtained from the corresponding author upon reasonable request.

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
