# Peer review of "Expression of IL-37 Induces a Regulatory T-Cell-like Phenotype and Function in Jurkat Cells"

_cells, 2022, doi:10.3390/cells11162565_

Round 1
Reviewer 1 Report
The manuscript entitled “IL-37 expression induces a regulatory T cell-like phenotype and function in Jurkat cells” investigates IL-37 expression in T lymphocytes with an innovative RNA-based methodology; the authors also describe the methodology to create a stable Treg-like cell line based on the commercial E6 Jurkat cells line lentivirus-transduced, demonstrating their Treg-like phenotype and function.
The manuscript is fluent, the data is organized comprehensively, and the topic is of interest to readers. The work can be improved in some points, especially regarding the discussion session.
Below are my suggestions to the authors:
- A substantial part of the discussion is devoted to comparing the results obtained in this study with those previously published by Rudloff and colleagues. The authors commented on the discrepancies and state that "Therefore, other factors may also be involved in the discrepancies, including IL-37 secretion amounts, cell fixation and permeabilization methodology and PBMC preparation methods"; however they mention neither the method of isolation of the PBMCs obtained in this study nor the method of fixation.
- A direct comparison with primary Treg cells should be presented, to evaluate whether, as for Kim (ref 6), "the suppressive activity (of the transduced Jurkat FOXP3 cells) was significantly lower than the primary Treg cells". Furthermore, I believe it is essential that the authors discuss the differences between the model presented in this paper and the one published by Kim and colleagues.
- I spent the entire reading wondering why the authors chose Jurkat cells and finally I found the answer in the conclusions. Probably this explanation should be anticipated in the introduction and, consequently, the conclusion rewritten.
- The text must be revised, with particular attention to the discussion, for typos and misphrases.
Author Response
"Please see the attachment."

Reviewer 2 Report
In this manuscript, authors attempt to produce a Treg-like cell line to facilitate Treg research. I think the premiss is promising, I think the results of the final figure indicate that this might be the case, but I have major concerns regarding the way the PrimeFlow assay data was analysed and displayed. Please find my recommendations below. As I am requesting some back-up data for the suppression assay, and due to the major change to the first figure, I am recommending major changes but urge authors to resubmit after revising the manuscript, as I do think the data is convincing with the alterations described below and is of interest to the scientific community.
Major
Figure 1
I do not agree at all with the order of the data and some of the data included in this figure – after panels A and B, panel E should follow. Ideally, this should show an overlay per cell type with activated and non-activated samples instead of the 2 histograms that are shown now, including unstained, to visually see whether a shift in MFI occurs. This is easier when the samples to be compared are in one plot. Furthermore, the way this is displayed now, it is impossible to quantify a percentage of cells expressing this cytokine or not (as in C and D). I am assuming this was done based on the isotype/unstained, but that is not really proper for this type of assay. Focussing on just the lymphoid cells to make this point - judging by the histograms, all cells in the unstimulated lymphocyte populations do not produce IL-37, where upon stimulation, there seems to be a shoulder visible for the CD8+ T cells and the B cells. I would ask the authors to provide overlay histograms per population of interest and place them as the third (set of) panel(s) in this figure so readers can themselves make a distinction whether there is an increase in IL37 MFI or not. Figure 1C and D need to be removed, as gating “cytokine positive” cells based on the unstained or secondary detection antibody/reagent (at least – that is what I think the gray filled histogram is – this is not mentioned in the figure legends by the way) is not at all appropriate; using a non-targeting probeset (i.e. against a mouse gene not expressed in human T cells) with the same fluorophore is a better control (doi: 10.1007/978-1-0716-0247-8_22). Also the discussion needs to be amended after removal of this data. I also had a look at the other paper authors mention where they measure IL37 via protein staining, but also there I am not super convinced with the way the gates are placed – in my opinion, the non-specific IgG control shows a lower MFI for the main population, and thus the positive gate there should have been a bit higher – more where the B cell population ends (this is nicely visible in panel C, where the IgG subset does not last until the end of the first peak of i.e. the CD8). If that type of analysis had been employed, the amount of IL37+ cells in resting would have been lower (there still would have been a difference for sure, I am not at all doubting that what they found is right), but more what one would expect.
Furthermore, it would maybe be of interest to plot IL-37 against a size parameter to see if the activation changes cell size, which can impact background (especially for flow cytometric RNA assessment this is of extreme importance) – this can be included in the supplementary file if needed.
Having said that, the analysis performed in Panel E is totally appropriate and substantiates the point authors are making – I would suggest to keep this panel and use that to draw conclusions.
Lastly – it would be ideal if authors could supply verification data of one celltype, e.g. Tregs via qPCR to validate the assay as this is a custom kit with a custom probeset.
Panel A – it seems like HLA-DR and CD11c are not compensated well, please double check. Similarly, for figure 1B, it would be more suitable to use a bi-exponential axis as half of the population is now missing in the CD56/CD16 plot.
Figure 4
For the suppression assay, it would be good if authors could supply an experiment in which Tresp were stimulated in the same conditions but in the presence of a gradient of IL-37 to verify whether it is the Jurkat cells that perform suppressive activity, or whether the presence of IL-37 is enough to inhibit T cell proliferation. Also, please write out Jurkat in the axis label on the right plot, there’s enough space.
Minor
Line 30 – remove the
Line 31 – enough = sufficient
Line 34 – please also add the advent of markers to identify stable and pure Tregs, such as GPA33 (doi: 10.4049/jimmunol.1901250)
Line 42, 55 – please add + after CD4
Line 47 – suggest to change to “suggesting that alternative pathways are required”
Line 48 – please rephrase Jurkats to Jurkat cells, and please do this consistently throughout the manuscript if applicable.
Line 56 – could might be better than would.
Line 61 and onward – I don’t completely agree with this statement – this is all dependent on the level of protein production. If all cells weakly produce this protein, it still could indicate that other cell types are more major contributors to the in vivo IL37 production. To say something conclusive about this, these cell types should be directly compared.
Line 80 – please add how the cells were cultured (humidified incubator at what temperature and oxygen/carbon dioxide pressure) and how often the cells were split/subcultured/medium was refreshed.
Line 81 – please add dilutions and fluorochromes for all antibodies as this is crucial information for combination with the PrimeFlow assay. Furthermore, in the prime-flow part, more antibodies (i.e. CD16/19) are mentioned. For all antibodies employed in this study, please state the manufacturer, clone, dilution and fluorophore employed. Alternatively, later on, a specific amount of antibody is mentioned, which could be re-calculated if authors state how many cells were stained in one sample.
Line 105 – change sups to supernatants and specific how the Takara kit was employed, or state “according to manufacturer’s instructions”.
Line 142 – there is a double period at the end of the sentence. I would furthermore propose to move the primer sets into a table for more clarity.
Line 144 – analysis of what? Please elaborate.
Line 155 – dilution = ratio
Line 169 – please elaborate as to why this type of assay was used – i.e. flow cytometric assessment of RNA is very suitable to detect mRNA of cytokine species of interest where antibody reagents are lacking/underperforming or when post-transcriptional regulatory mechanisms influence cytokine production. To support this, several good reviews exist regarding the use of Flow-FISH in T cells to measure cytokine production (e.g. 10.1615/CritRevImmunol.2018025938 or 10.1615/CritRevImmunol.2017018316) which could be referenced.
Figure 1E – perhaps it would be easier to distinguish the two activation types if activated cells were a different symbol, i.e. a triangle or circle.
Figure 1E – please rename NKs to NK cells
Line 203 – started transducing = transduced
Figure 2B – why did authors now switch from PrimeFlow to qPCR? It would have made more sense and strengthened the manuscript if authors sticked to the same type of readout and used qPCR to confirm the findings of the PrimeFlow assay.
Line 260 – It seems like you forgot to add a citation.
Line 268 – please correct the sentence, I think it should read “only”
Author Response
"Please see the attachment."

Reviewer 3 Report
In the manuscript Cells-1768677 by Osborne et al, authors have shown the development of Treg phenotype and functions upon ectopic expression IL-37. Authors have used different experimental assays including flow cytometry, flow RNA assay, qPCR, western blotting and T cell suppression assay.
Authors also have acknowledged a previously published report by Rudloff et al and discussed the discrepancy.
Data is statistically analyzed, and the presented results support the conclusions. Overall, manuscript is well written and explained. The findings are useful for the field.
I have some minor points to address.
Figure 2. Panel C. Please provide the number of replicates and error bars to the graph. Panel B. Please define the Y-axis value.
Figure 3. Panel A. what the Y-axis values? Are they fold change? Please define.
Figure 4. First panel, x-axis label is missing. Second panel, please define the black and grey bars.
Author Response
"Please see the attachment."

Round 2
Reviewer 1 Report
The authors answered the objections and resolved the critical issues. The article in the revised version is worthy for publication.
Reviewer 2 Report
Thank you for taking my questions and queries seriously and adressing most of them. I think the manuscript has improved significantly and is now suitable for publication. Please note - in the answers to my queries, you stated you cited the GPA 33 paper in the introduction, but the reference is missing. Not having it there is fine, but then also the specific mention of GPA33 should be removed I think. This can be fixed upon author correction stage. Congratulations on an interesting study.